# Effects of a Mixed Emissions Control Policy on the Manufacturer’s Production and Carbon Abatement Investment Decisions

**DOI:** 10.3390/ijerph192013472

**Published:** 2022-10-18

**Authors:** Fei Wang, Dalin Zhang

**Affiliations:** 1National Tax Institute of STA, Yangzhou 225007, China; 2Department of Computer Science, Aalborg University, 9220 Aalborg, Denmark

**Keywords:** mixed emissions control policy, carbon tax, dynamic reward-punishment mechanism, production decision, abatement investment decision

## Abstract

Considering the consumers’ environmental awareness, a mixed emissions control policy with carbon tax and a dynamic reward-punishment mechanism for carbon abatements was introduced to explore the manufacturer’s low-carbon production issues. The results showed that: (1) Under a given mixed emissions control policy, a higher government pre-determined abatement target cannot positively encourage manufacturers’ carbon abatement behaviors. However, a stricter emissions control policy is environmentally beneficial only when the government pre-determined abatement target exceeds a certain threshold. (2) Reducing the carbon abatement cost and enhancing the consumers’ environmental awareness would always benefit manufacturers’ low-carbon production, but both approaches benefit the environment only when the government pre-determined abatement target is below a certain threshold. (3) Under a mixed emissions control policy of social welfare maximization, the reward-punishment coefficient positively correlates with the government’s optimal pre-determined abatement target, and the effect of the carbon tax rate on that is closely related to the carbon emissions of the unit product. More importantly, imposing a carbon tax or raising the tax rate and adopting a reward-punishment mechanism or raising the reward-punishment coefficient can effectively encourage manufacturers’ carbon abatement investment behaviors. However, they have nothing but a negative effect on manufacturers’ excessive abatement levels.

## 1. Introduction

Global warming has become a global challenge, and it is urgent to reduce emissions of carbon dioxide and other greenhouse gases. Many countries have set clear carbon abatement targets for the increasingly severe emission reduction situation. For example, the U.S. pledged to reduce greenhouse gas emissions by about half by 2030 compared to 2005, and the U.K. also announced a 68% reduction in greenhouse gas emissions by 2030 compared to 1990 [1,2]. In China, the government has further emphasized a reduction in CO_2_ emissions per unit of GDP by more than 65% by 2030 compared to 2005, based on the target of carbon peaking and carbon neutrality [3]. To achieve these action goals, several emission control policies have been promulgated, such as mandatory carbon capacity, carbon tax, emission trading mechanism, and carbon offsets [4,5]. Among them, the carbon tax has played a positive role in promoting low-carbon development [6,7] and has been implemented in many countries, such as Ireland, Sweden, and the U.S. [8,9].

However, some studies show that the carbon tax still has some limitations in controlling emissions [10,11] and also results in higher operating costs and lower total profits for manufacturers and even slower economic growth [12]. Therefore, the government introduced the reward-punishment mechanism to guide and regulate manufacturers’ carbon abatement behaviors [13]. In practice, in addition to some static reward-punishment mechanisms, a number of dynamic reward-punishment mechanisms for carbon emissions and carbon abatements have also been proposed to further promote the low-carbon development. For instance, in 2019, the ‘Regulation (EU) 2019/631 of the European Parliament and of the Council’ issued by the European Union set CO_2_ emission performance standards for new passenger cars and new light commercial vehicles. Those manufacturers who are unqualified for the new rule face fines based on excess emissions and the number of newly registered vehicles [14]. At the end of 2020, Volkswagen was fined up to EUR 275 million for failing to meet the corresponding abatement target [15]. In China, local governments have also introduced similar reward-punishment mechanisms for energy saving and carbon abatement. For example, in 2021, the ‘Implementation Scheme of Financial Reward and Subsidy Mechanism for Ecological Civilization Construction in Dongying City’ was introduced by Dongying in Shandong province. This implementation scheme clearly proposed a reward-punishment mechanism for carbon abatements, giving certain cash rewards to manufacturers based on the actual abatement ratio above the government’s pre-determined target [16]. Under the dynamic reward-punishment mechanisms, the targeted number of emissions or the targeted number of abatements is not fixed and is closely related to the manufacturer’s total output. Then, the corresponding rewards or penalties are given according to manufacturers’ actual carbon emission or carbon abatement gaps and the reward-punishment coefficient. This paper takes the dynamic reward-punishment mechanism for carbon abatements under the carbon tax environment as the mixed emissions control policy.

The emissions control policy prompts manufacturers to carry out low-carbon investments actively, thereby improving the product abatement level [17,18]. In addition, more consumers are showing a stronger environmental awareness and willingness to purchase low-carbon products [9,19,20]. For instance, a global survey conducted by Accenture shows that more than 80% of consumers pay attention to the environmental property of products when purchasing them [21]. In addition, the Research Institute for Ecological-civilization of Chinese Academy of Social Sciences found that nearly 90% of Chinese consumers are willing to choose products from environmentally friendly enterprises [22]. Therefore, the growing concept of low-carbon consumption has become a new market driver, encouraging manufacturers to positively carry out carbon abatement investment activities [23]. Chinese manufacturers such as Gree and Haier have been pushing new low-carbon products to enhance their environment-friendly image and the competitive advantage of their products [24]. However, exploiting low-carbon technologies will inevitably raise manufacturers’ production and operation costs [4]. Then, whether the incremental revenue brought by carbon abatement can compensate for the additional investment cost will be focused on by manufacturers. Furthermore, from the government’s perspective, the pursuit of maximum social welfare by considering the interests of manufacturers, consumers, and the environment will be the initial motivation of the emissions control policy design and optimization.

In view of the aforementioned analysis of realistic background, this paper aimed to explore the manufacturer’s low-carbon production issues under the dynamic reward-punishment mechanism for carbon abatements issued by the government considering the carbon tax policy. The design and optimization of the mixed emission control policy was investigated with the objective of maximizing social welfare. Some research questions are answered in this paper: (1) How does the dynamic reward-punishment mechanism for carbon abatements affect manufacturers’ low-carbon operations under a mixed emissions control policy? (2) Under what conditions can the mixed emissions control policy better promote manufacturers’ abatement investments and low-carbon production or be more viable for the environment? (3) Under the objective of maximizing social welfare, how does the government determine the optimal pre-determined abatement target, and how does the optimal mixed emissions control policy affect manufacturers’ carbon abatement behaviors?

To address the above issues, firstly, a profit maximization model was constructed under a given mixed emissions control policy. The manufacturer determines the optimal carbon abatement investment level and production quantity of low-carbon products based on the known carbon tax rate, the reward-punishment coefficient, and the government pre-determined abatement target. Then, a social welfare maximization model is constructed under the established carbon tax environment, and the government determines the optimal pre-determined abatement target based on the manufacturer’s decision feedback. Furthermore, through theoretical analysis and numerical analysis, this paper explored the effect of the mixed emissions control policy and consumers’ environmental awareness of the manufacturer’s carbon abatement investment decision and production decision and the environment. Finally, we further investigated the decision-making process of the government’s optimal pre-determined abatement target and the manufacturer’s carbon abatement behaviors under the optimal mixed emissions control policy.

The remainder of this paper is organized as follows. Section 2 is devoted to a review of the related literature. The case study and the key assumptions are described in detail, and the methods are proposed in Section 3. The profit maximization model formulation and theoretical analysis are presented in Section 4, and Section 5 explores the government’s optimal pre-determined carbon abatement target by maximizing social welfare. Finally, conclusions and future research are provided in Section 6.

## 2. Literature Review

So far, a great deal of academic research has been addressed on manufacturers’ abatement investment issues. Generally, two crucial drivers affect manufacturers’ carbon abatement investment behaviors: consumers’ environmental awareness and the emission control policy. Therefore, this paper reviews the literature from these two aspects.

Considering that consumers are willing to pay higher prices for low-carbon products, Zhu and He [25] studied the effect of the supply chain structure, green product types, and competition patterns on a supply chain’s greening effort level. Du et al. [26] analyzed low-carbon abatement behaviors at the supply chain level when each member firm independently determines its effort level. Introducing consumer low-carbon preferences to the differential game model, Liang and Liu [27] studied the equilibrium strategy of a two-echelon supply chain under centralized and decentralized decision-making conditions and discussed the effect of low-carbon preference on emission reduction behaviors. Zhang et al. [28] explored the effect of retailers’ cost-sharing, fairness concerns, and advertising investment on the supply chain members’ decisions by considering both consumers’ environmental awareness and product abatement rate. Considering the consumer low-carbon preference, Zhang et al. [29] explored the impact of carbon emission reduction on supply chain operations and financing decisions under three different strategies (i.e., bank loan financing, equity financing, and hybrid financing). Utilizing discrete choice experiments, Mazzocchi et al. [30] found that consumers’ environmental awareness could better facilitate the Italian pork industry to explore carbon abatement techniques to reduce carbon emissions. However, none of the above literature considered the effect of the government’s emissions control policies.

Due to the importance of low-carbon development, various states and local governments have formulated several emission control policies to promote enterprises’ carbon abatement investments, including mandatory and incentive-punishment policies. Among them, many scholars have focused on the effect of carbon tax on abatement investment behaviors regarding mandatory policies. For example, considering carbon tax and take-back legislation, Ding et al. [7] examined remanufacturing and emission reduction strategies in monopolistic and competitive environments. Alegoz et al. [31] comparatively analyzed production and emission reduction investment decisions in a pure manufacturing system and a hybrid manufacturing/remanufacturing system. Introducing a differentiated carbon tax policy for new and remanufactured products, Wang and Wang [32] mainly explored the effect on manufacturing/remanufacturing and emission reduction decisions. For centralized and decentralized closed-loop supply chains, Luo et al. [33] addressed manufacturing and remanufacturing decision issues with and without abatement technology investments under the carbon tax policy. Wei and Huang [34] discussed the cash flow, the inventory risk allocation, and the impacts of carbon tax on greening technology investment, production volume, and order quantity decisions under different contracts (i.e., advance purchase discount and prepayment-based option). However, the above literature ignored the effect of consumers’ environmental awareness on the demand for low-carbon products. Considering consumers’ low-carbon preference, Yang and Chen [35] analyzed the effect of retailer revenue sharing and cost sharing under carbon tax policy on the manufacturer’s carbon abatement efforts and profitability. Focusing on different power structures, Huang and Zhang [36] studied a supply chain’s optimal emission reduction and pricing decisions. Zhang et al. [4] explored the production/pricing and emission reduction decisions of the manufacturer by introducing a progressive carbon tax policy. Based on differentiated carbon tax rates faced by manufacturers and overseas suppliers, Yang et al. [37] mainly focused on exploring the effect of carbon tax and consumers’ environmental awareness on optimal in-house production and outsourcing decisions for multinational manufacturers with different emission reduction technologies.

As for incentive-punishment policies, more literature focused on the effect of government subsidies on carbon abatement investment decisions [5,12,38,39]. There are also a few studies concerned with the relationship between the government reward-punishment mechanism and low-carbon abatement activities. For instance, Wang et al. [40] discussed the operation issues of reverse supply chains under different policy combinations, integrating carbon emissions and the reward-punishment mechanism of the recycling rate. Wang et al. [41] considered the government’s reward-punishment for manufacturers’ carbon emissions and conducted research on differential pricing decisions for new and remanufactured products in a closed-loop supply chain. Villicaña-García et al. [42] proposed an economic incentive-punishment scheme based on greenhouse gas emission limits and further studied strategic planning issues of the energy supply chain, including fossil fuels and biofuels. Based on an improved tiered reward-punishment carbon trading policy, Zhang et al. [43] constructed a two-stage benefit sharing model for a mixed renewable energy system. It can be found that when examining the effect of incentive-punitive policies on low-carbon activities, mandatory policies are rarely considered comprehensively, nor do they integrate carbon abatement investments and the corresponding effect on the demand for low-carbon products. More importantly, the existing literature does not involve the reward-punishment mechanism for manufacturers’ carbon abatement gaps and the design and optimization of corresponding policies.

A summary of relevant literature is presented in Table 1 to compare emission control policies, decision variables, consumers’ environmental awareness, and decision-makers from these previous studies. This paper mainly contributes to the literature in the following three aspects. First, to the best of our knowledge, this paper was the first study on the reward-punishment mechanism based on the government’s pre-determined abatement target and the manufacturer’s actual product low-carbon level. Second, joint production and abatement investment decisions under a mixed emissions control policy, i.e., carbon tax and dynamic reward-punishment mechanisms for carbon abatements, were explored. Third, we also investigated the effect of the mixed emissions control policy on the environment. We also examined the decision-making process of the government’s optimal pre-determined abatement target by maximizing social welfare. Through theoretical analysis and numerical analysis, some managerial insights and policy implications are provided for the manufacturer’s low-carbon production and the government’s policy design, respectively.

## 3. Problem Description and Assumptions

### 3.1. Problem Description and Symbol Instruction

This paper considers a monopoly manufacturer engaged in the production-sale of low-carbon products. Due to the popularization of environmental protection concepts, consumers are willing to pay higher prices for low-carbon products. As the advocate of low-carbon development, the government promotes the manufacturer to invest in low-carbon technologies by implementing the carbon tax policy. Meanwhile, in order to further guide and regulate the manufacturer’s carbon abatement behaviors, the government also adopts a dynamic reward-punishment mechanism for carbon abatements. That is, under the government pre-determined abatement target, the manufacturer would be rewarded or punished according to carbon abatement gaps and the total output of low-carbon products. The government pre-determined carbon abatement target represents the strictness of the mixed emissions control policy. Under the mixed emissions control policy, the manufacturer decides on the optimal abatement investment level and production quantity of low-carbon products to maximize profit. It should be noted that the optimization models formulated in this paper are in a single-period decision environment, which can be regarded as a steady-state period in an infinite horizon. Similar settings can be found in the previous literature [7,32]. All parameters and decision variables involved in our models are shown in Table 2.

### 3.2. Assumptions

To better understand our model, the key assumptions are shown as follows:

**Assumption** **1.**
*Following [35] and Cao et al. [44], the product demand is sensitive to the sales price, and consumers would pay the higher price for low-carbon products. In addition, it should be noted that the production quantity is equal to the product demand, and both the market size and the price sensitivity coefficient are 1; a similar assumption can be found in Liu et al. [45]. Then, the demand function of low-carbon products can be expressed as q_n_ = 1 − p_n_ + λτ. Thus, the inverse function can be derived as p_n_ = 1 − q_n_ + λτ.*


**Assumption** **2.**
*The emission reduction activity can be regarded as a one-off investment, and the higher the emission reduction investment, the greater the emission reduction cost. Then, following [7,32], the carbon abatement cost is assumed to be a quadratic function kτ^2^/2.*


**Assumption** **3.**
*Referring to [32,44], the government is the leader and is committed to maximizing social welfare through pre-determining the carbon abatement target under the given carbon tax rate and reward-punishment coefficient. It should be noted that the higher the government pre-determined abatement target, the stricter the mixed emissions control policy. The manufacturer is the follower and is committed to maximizing its total profit and determines the abatement investment level and production quantity successively. Finally, the sequence of participants’ decisions is shown in Figure 1.*


**Assumption** **4.**
*Referring to Shen et al. [46] and Ding et al. [47], it is assumed that two classes of participants are risk-neutral and there is no information asymmetry, i.e., both the manufacturers and the government share information.*


### 3.3. The Methods

As mentioned above, the government is the leader, and the manufacturer is the follower. The sequence of participants’ decisions is depicted in Figure 1. At first, the government decides the pre-determined abatement target *ϕ* under the given carbon tax rate *t*_0_ and reward-punishment coefficient *η*. In addition, under a given mixed emission control policy, the manufacturer sets the carbon abatement investment level *τ*, and then decides the production quantity/sales price (*q_n_*/*p_n_*) of low-carbon products. Finally, the product demand is satisfied at a given price. Thus, the game model is built between the government and the manufacturer and is solved by using the backward induction method.

## 4. Model Construction and Analysis

### 4.1. Manufacturer’s Optimal Decisions

In this subsection, we mainly discuss how the manufacturer decides the abatement investment level and production quantity of low-carbon products when the mixed emission control policy is given. Additionally, the effects of some critical parameters, such as the carbon abatement cost coefficient *k* and the consumer low-carbon preference coefficient λ, and the government pre-determined abatement target *ϕ*, carbon tax rate *t*_0_, and reward-punishment coefficient *η* on the manufacturer’s optimal operation decisions are further analyzed. According to the previous problem description, the manufacturer’s profit maximization model function is shown in Equation (1). All proof is presented in Appendix A:(1)$$\pi_{m}=(1-q_{n}+\lambda \tau )q_{n}-t_{0}e_{n}q_{n}(1-\tau )+\eta e_{n}q_{n}(\tau -\phi )-\frac{1}{2}k\tau^{2}$$
where the first term represents the manufacturer’s sales revenue of low carbon products, the second term represents the carbon tax cost, the third term represents the incentive incomes or penalty payouts of carbon abatement gaps, and the last term represents the carbon abatement cost of the manufacturer.

**Lemma** **1.***Under a given emissions control policy, the total profit function is jointly concave in τ and q_n_, and the expressions of the manufacturer’s optimal abatement investment level, the optimal production quantity of low-carbon products, and the corresponding optimal sales price are*$\tau^{*}=\frac{H_{0}(1-t_{0}e_{n}-\eta \cdot \phi e_{n})}{2k-H_{0}^{2}}$, $q_{n}^{*}=\frac{k(1-t_{0}e_{n}-\eta \cdot \phi e_{n})}{2k-H_{0}^{2}}$*, and*$p_{n}^{*}=\frac{\left(2k-H_{0}^{2})-(k-\lambda H_{0})(1-t_{0}e_{n}-\eta \cdot \phi e_{n}\right)}{2k-H_{0}^{2}}$*, where*$\phi <\frac{1-t_{0}e_{n}}{\eta e_{n}}=\phi_{0}$*,*$k>\frac{H_{0}^{2}+H_{0}(1-t_{0}e_{n}-\eta \cdot \phi e_{n})}{2}$*, and*$H_{0}=\lambda +t_{0}e_{n}+\eta e_{n}$*.*

According to Lemma 1, the effects of carbon abatement cost coefficient *k*, consumer low-carbon preference coefficient *λ*, government pre-determined abatement target *ϕ*, carbon tax rate *t*_0_, and reward-punishment coefficient *η* on the optimal abatement investment and production decisions are demonstrated in Propositions 1–4, respectively.

**Proposition** **1.***(1)*$\frac{\partial \tau^{*}}{\partial k}<0$*;*$\frac{\partial q_{n}^{*}}{\partial k}<0$*; if*  
$\lambda <(t_{0}+\eta )e_{n}$*, then*  
$\frac{\partial p_{n}^{*}}{\partial k}>0$*, otherwise,*  
$\frac{\partial p_{n}^{*}}{\partial k}<0$*; (2)*
$\frac{\partial \tau^{*}}{\partial \lambda }>0$
*;*
$\frac{\partial q_{n}^{*}}{\partial \lambda }>0$*; if*  
$k>\frac{H_{0}^{2}(t_{0}+\eta )e_{n}}{2\lambda }$*, then*  
$\frac{\partial p_{n}^{*}}{\partial \lambda }>0$*, otherwise,*  
$\frac{\partial p_{n}^{*}}{\partial \lambda }<0$*.*

Proposition 1 illustrates that the rising abatement cost coefficient *k* negatively affects the manufacturer’s carbon abatement investment level and production quantity of low-carbon products. This is consistent with the results of most existing studies. However, if the consumer low-carbon preference coefficient *λ* is below a certain threshold (*λ* < (*t*_0_ *+ η*)*e_n_*), the manufacturer would raise the sales price of low-carbon products with the increase in the carbon abatement cost coefficient to cover the loss caused by the declined output. This also means the transfer of carbon abatement costs to consumers. If the consumer low-carbon preference coefficient is large enough (*λ* > (*t*_0_ *+ η*)*e_n_*), the manufacturer may reduce the sales price of low-carbon products with the increase of the abatement cost coefficient. This will effectively avoid a larger decline in product demand caused by the falling carbon abatement investment level, thereby ensuring the profit maximization of the manufacturer. In addition, the rising consumer low-carbon preference coefficient always prompts the manufacturer to improve the carbon abatement investment level and production quantity of low-carbon products. This is also consistent with the actual situation. Furthermore, the manufacturer raises the sales price of low-carbon products when the carbon abatement cost coefficient is larger. This will cover the higher carbon abatement cost. Otherwise, the manufacturer may reduce the sales price as the increase of the low-carbon preference coefficient. Consequently, small profits but a quick turnover will help the manufacturer better achieve a win-win target of economic and environmental benefits.

**Proposition** **2.***(1)*$\frac{\partial \tau^{*}}{\partial \phi }<0$*;*$\frac{\partial q_{n}^{*}}{\partial \phi }<0$*; (2) if*  
$k>\lambda H_{0}$*, then*  
$\frac{\partial p_{n}^{*}}{\partial \phi }>0$*, otherwise,*  
$\frac{\partial p_{n}^{*}}{\partial \phi }<0$*.*

Proposition 2 shows that a stricter emissions control policy (i.e., a higher government pre-determined abatement target) does not necessarily provide effective incentives for the manufacturer to improve carbon abatement investment level. Instead, it will increase the manufacturer’s cost burden and reduce the market demand for low-carbon products, which is not conducive to low-carbon production and the sustainable development of the manufacturer. Moreover, to obtain a higher total profit, the manufacturer raises the sales price facing an increasingly stringent emission control policy if the carbon abatement cost coefficient is larger or the consumer low-carbon preference coefficient is relatively low (*k > λH*_0_). The main reason is because when the abatement cost coefficient is high or the low-carbon preference coefficient is low, the magnitude of the decline in the carbon abatement investment level and production quantity caused by the rising government pre-determined abatement target are relatively small. Then, the manufacturer can appropriately increase the sales price of low-carbon products to maximize the total profit. However, if the abatement cost coefficient is low or the low-carbon preference coefficient is high (*k < λH*_0_), the more substantial the decline in the carbon abatement investment level and production quantity will occur with the increase of government pre-determined abatement target. Thus, the manufacturer must stimulate the market demand by reducing the sales price of low-carbon products to compensate for the possible rising cost or falling revenue. To sum up, if the abatement cost coefficient is high or the low-carbon preference coefficient is low, the manufacturer can maximize the total profit by raising the sales price of low-carbon products when facing a stricter emissions control policy. When the abatement cost coefficient is low or the low-carbon preference coefficient is high, reducing the sales price of low-carbon products is often more beneficial to the manufacturer. It also means that both reducing the carbon abatement cost and increasing consumers’ environmental awareness can effectively avoid the excessive transfer of operating costs by manufacturers under increasingly stringent emission control policies.

According to the manufacturer’s decision feedback, the government’s pre-determined carbon abatement target should satisfy the following equation:$$\phi =\tau^{*}=\frac{\left(\lambda +t_{0}e_{n}+\eta e_{n})(1-t_{0}e_{n}-\eta \cdot \phi e_{n}\right)}{2k-{\left(\lambda +t_{0}e_{n}+\eta e_{n}\right)}^{2}}=\frac{H_{0}(1-t_{0}e_{n}-\eta \cdot \phi e_{n})}{2k-H_{0}^{2}}$$

Then, we can obtain an optimal pre-determined carbon abatement target in a profit-maximizing situation, i.e., $\phi^{*}=\frac{H_{0}(1-t_{0}e_{n})}{2k-H_{0}^{2}+\eta e_{n}H_{0}}$. Then, if 0 < *ϕ* < *ϕ**, the manufacturer can easily meet the government pre-determined abatement target and obtain incentive income by carrying out carbon abatement investment activities. If *ϕ** < *ϕ* < *ϕ*_0_, a certain amount of penalty payouts will occur due to the insufficient carbon abatement. If *ϕ* > *ϕ*_0_, the extremely strict emissions control policy would force heavy-emission manufacturers to withdraw from the market.

**Proposition** **3.***If*  
$0<\phi <\phi_{2}$*, then* 
$\frac{\partial \tau^{*}}{\partial t_{0}}>0$
*and*  
$\frac{\partial q_{n}^{*}}{\partial t_{0}}>0$*; if*  
$\phi_{2}<\phi <\phi_{1}$*, then*  
$\frac{\partial \tau^{*}}{\partial t_{0}}>0$
*and*  
$\frac{\partial q_{n}^{*}}{\partial t_{0}}<0$*; if*  
$\phi_{1}<\phi <\phi_{0}$*, then*  
$\frac{\partial \tau^{*}}{\partial t_{0}}<0$
*and*  
$\frac{\partial q_{n}^{*}}{\partial t_{0}}<0$*.*

Proposition 3 implies that the effect of carbon tax on the manufacturer’s carbon abatement investment and low-carbon production decisions mainly depends on the strictness of the emission control policy. Specifically, when the government pre-determined abatement target is relatively low (0 < *ϕ* < *ϕ*_2_), the rising carbon tax rate improves the manufacturer’s carbon abatement investment level and production quantity of low-carbon products. When the pre-determined abatement target is further raised (*ϕ*_2_ < *ϕ* < *ϕ*_1_), a rising carbon tax rate still induces the manufacturer to improve the carbon abatement investment level but fails to increase the production quantity of low-carbon products. Finally, when the emissions control policy is stricter (*ϕ*_1_ < *ϕ* < *ϕ*_0_), the rise in the carbon tax rate can hardly motivate the manufacturer to further improve the carbon abatement investment level. It instead increases the cost burden and ultimately reduces the market demand for low-carbon products, which is consistent with Proposition 2. This also shows that raising the tax rate is not necessarily beneficial to low-carbon production and the sustainable development of the manufacturer under a more stringent mixed emission control policy. The main reason is that a stricter emission control policy (*ϕ* > *ϕ*_2_) not only brings a lower carbon abatement investment level but also leads to a continuous decline in the growth magnitude of the abatement investment level caused by the rising carbon tax rate. Under these circumstances, the manufacturer must decrease the production quantity of low-carbon products to reduce the carbon tax cost effectively. Until the emissions control policy becomes further stringent (*ϕ*_1_ < *ϕ* < *ϕ*_0_), the government will reduce the tax rate to promote the manufacturer’s carbon abatement investment activities.

**Proposition** **4.***If*  
$0<\phi <\phi_{4}$*, then* 
$\frac{\partial \tau^{*}}{\partial \eta }>0$
*and*  
$\frac{\partial q_{n}^{*}}{\partial \eta }>0$*; if*  
$\phi_{4}<\phi <\phi_{3}$*, then*  
$\frac{\partial \tau^{*}}{\partial \eta }>0$
*and*  
$\frac{\partial q_{n}^{*}}{\partial \eta }<0$*; if*  
$\phi_{3}<\phi <\phi_{0}$*, then*  
$\frac{\partial \tau^{*}}{\partial \eta }<0$
*and*  
$\frac{\partial q_{n}^{*}}{\partial \eta }<0$*.*

Similar to Proposition 3, Proposition 4 indicates that under the stricter emission control policy, the positive incentive effect of the reward-punishment mechanism on the manufacturer’s carbon abatement investment level and the production quantity of low-carbon products appears to be weaker, until the negative effect occurs. Remarkably, when the government pre-determined abatement target is higher than a certain threshold (*ϕ* > *ϕ*_3_), the manufacturer will never reach it (*ϕ*_3_ *> ϕ**). Therefore, a higher reward-punishment coefficient further increases the manufacturer’s cost burden and ultimately reduces the carbon abatement investment level and production quantity of low-carbon products. From Proposition 3 and Proposition 4, it can be found that the carbon tax rate or the reward-punishment coefficient has the opposite effect on the manufacturer’s low-carbon operations when comparing the looser and stricter emission control policies. Fortunately, changes in both of the above two factors in different directions can promote the manufacturer to carry out carbon abatement activities and simultaneously protect production activities of low-carbon products. More specifically, under a relatively loose emissions control policy, a higher carbon tax rate or reward-punishment coefficient is conducive to the manufacturer’s carbon abatement investment and low-carbon production activities. Otherwise, the government must reduce the carbon tax rate or the reward-punishment coefficient to guide the manufacturer to better achieve a win-win situation of economic and environmental benefits.

### 4.2. Effects on the Environment

This subsection analyzes the effect of some parameters mentioned above on the environment. Referring to [7,32], the manufacturer’s total carbon emissions are used to represent the effect on the environment, as shown in Equation (2).
(2)$$E_{m}=e_{n}q_{n}^{*}(1-\tau^{*})=\frac{ke_{n}(1-t_{0}e_{n}-\eta \cdot \phi e_{n})}{2k-H_{0}^{2}}-\frac{ke_{n}H_{0}{\left(1-t_{0}e_{n}-\eta \cdot \phi e_{n}\right)}^{2}}{{\left(2k-H_{0}^{2}\right)}^{2}}$$

**Proposition** **5.***If*  
$0<\phi <\phi_{5}$*, then*  
$\frac{\partial E_{m}}{\partial k}>0$
*and*  
$\frac{\partial E_{m}}{\partial \lambda }<0$*; if*  
$\phi_{5}<\phi <\phi_{1}$*, then*  
$\frac{\partial E_{m}}{\partial k}>0$
*and*  
$\frac{\partial E_{m}}{\partial \lambda }>0$*; if*  
$\phi_{1}<\phi <\phi_{0}$*, then*  
$\frac{\partial E_{m}}{\partial k}<0$
*and*  
$\frac{\partial E_{m}}{\partial \lambda }>0$*.*

Proposition 5 shows that, although the declining abatement cost coefficient always increases the manufacturer’s carbon abatement investment level and production quantity of low-carbon products, it can also lead to lower total carbon emissions only when the government pre-determined abatement target is below a certain threshold (*ϕ* < *ϕ*_1_). Similarly, as shown in Proposition 1, the stronger consumer low-carbon preference is always conducive to improving the manufacturer’s carbon abatement investment level and production quantity of low-carbon products. At this time, if the government pre-determined abatement target is below a certain threshold (*ϕ* < *ϕ*_5_), then the rising low-carbon preference coefficient leads to lower total carbon emissions. This is mainly because, under the relatively loose emission control policy, the growth rate of the carbon abatement investment level is higher than that of the production quantity due to the decrease of the abatement cost coefficient or the increase of the low-carbon preference coefficient. Otherwise, when the emissions control policy is stricter (*ϕ*_1_ < *ϕ* < *ϕ*_0_), the increased economic benefits brought by the lower abatement cost or the stronger environmental awareness come at the cost of more heavy environmental damage. Therefore, under environmental protection pressure, the manufacturer should choose effective carbon abatement approaches according to the strictness of the emissions control policy, such as reducing the carbon abatement cost or enhancing the consumer environmental awareness. For instance, when the government pre-determined abatement target satisfies *ϕ*_5_ < *ϕ* < *ϕ*_1_, reducing the carbon abatement cost is more beneficial to economic and environmental benefits for the manufacturer. Remarkably, when the government pre-determined abatement target is below a certain threshold (*ϕ* < *ϕ*_5_), the above two approaches are both conducive to realizing the win-win target of economic and environmental benefits. However, when the emission control policy is much stricter (*ϕ* > *ϕ*_1_), neither approach is environmentally friendly. This also implies that the much stricter emission control policy restricts the options available to the manufacturer’s low-carbon development.

**Proposition** **6.***If*  
$0<\phi <\phi_{2}$*, then*  
$\frac{\partial E_{m}}{\partial \phi }>0$*; if*  
$\phi_{2}<\phi <\phi_{0}$*, then*  
$\frac{\partial E_{m}}{\partial \phi }<0$*.*

From Proposition 6, it can be found that the manufacturer’s total carbon emissions reach their peak when the government pre-determined abatement target is equal to a certain value (*ϕ* = *ϕ*_2_). That is to say, this government pre-determined abatement target is meaningless from an environmental perspective at this time. Overall, the effect of the strictness of the emission control policy on the total carbon emissions also depends on the range of the government pre-determined abatement target. When 0 < *ϕ* < *ϕ*_2_, as the government pre-determined abatement target decreases, the manufacturer’s total carbon emissions decline, while the carbon abatement investment level and production quantity of low-carbon products increase. This implies that within this range, a looser emission control policy can promote the manufacturer’s carbon abatement behaviors, thereby reducing environmental damage. Conversely, when *ϕ*_2_ < *ϕ* < *ϕ*_0_, as the government pre-determined abatement target decreases, the manufacturer’s total carbon emissions increase accompanied by an increased carbon abatement investment level and production quantity of low-carbon products. This means that, within this range, the rising pre-determined carbon abatement target is more conducive to reducing environmental damage. Therefore, for the government, a loose emission control policy is often more beneficial to the manufacturer’s sustainable development. However, whether the stricter or looser emissions control policy benefits the environment needs to refer to the threshold value of the government pre-determined abatement target that results in the manufacturer’s highest carbon emissions.

**Proposition** **7.***(1)*  
$\frac{\partial E_{m}}{\partial t_{0}}<0$*; (2) if*  
$\phi <\phi_{2}$*, then*  
$\frac{\partial E_{m}}{\partial \eta }<0$*.*

Proposition 7 illustrates that, although the effect of the carbon tax rate on the manufacturer’s carbon abatement investment level and production quantity of low-carbon products also depends on the government pre-determined abatement target, levying a carbon tax or raising the carbon tax rate is always beneficial to controlling the manufacturer’s total carbon emissions. As for the dynamic reward-punishment mechanism, only when the government pre-determined abatement target is below a certain value (*ϕ* < *ϕ*_2_) can it ensure that adopting a reward-punishment mechanism or raising the reward-punishment coefficient always has a positive effect on the environment. This is mainly because, as stated in Proposition 4, a higher reward-punishment coefficient is more conducive to improving the manufacturer’s carbon abatement investment level under the looser emission control policy. It also implies that, compared with the reward-punishment mechanism, the role of carbon tax is more direct and effective in controlling the total carbon emissions of the manufacturer.

## 5. Government’s Optimal Decisions

As a leader, the government designs and adjusts the emissions control policy according to the manufacturer’s decision feedback. This section mainly explores how the government decides the optimal pre-determined abatement target (namely, the strictness of the mixed emissions control policy) under the given carbon tax rate and reward-punishment coefficient to maximize social welfare. Meanwhile, the optimal low-carbon production activities (i.e., carbon abatement investment level, production quantity of low-carbon products, and the corresponding excessive abatement level) of the manufacturer are investigated under the emission control policy of maximizing social welfare. Therefore, the excessive abatement level of the manufacturer is equal to the difference of the manufacturer’s optimal carbon abatement investment level and the government’s optimal pre-determined abatement target. This indicator represents the degree to which the manufacturer meets the government’s carbon abatement requirements. If it is positive, the manufacturer will obtain the incentive benefits for excessive carbon abatements, otherwise, a fine is imposed. Moreover, the social welfare *π_g_* is defined as the sum of the manufacturer’s total profit *π_m_* and the consumer surplus *π_c_* minus the environmental damage *π_e_* and the government expenditure *f_g_*. Among them, the consumer surplus and environmental damage are obtained referring to [19,32], while government expenditure is the difference between the incentive expenditure and carbon tax income. Therefore, the social welfare maximization model function can be written as follows:(3)$$\pi_{g}=\pi_{m}+\pi_{c}-\pi_{e}-f_{g}=(1-q_{n}^{*}+\lambda \tau^{*})q_{n}^{*}-\frac{1}{2}k\tau^{*2}+\frac{q_{n}^{*2}}{2}-\mu e_{n}q_{n}^{*}(1-\tau^{*})$$

**Lemma** **2.**
*Under the given carbon tax rate and reward-punishment coefficient, the social welfare function is a concave function in ϕ, and the expression of the optimal pre-determined abatement target is:*

$\phi^{**}=\frac{1-t_{0}e_{n}}{\eta e_{n}}-\frac{\left(1-\mu e_{n})(2k-H_{0}^{2}\right)}{\eta e_{n}(k+H_{0}-2\lambda H_{0}-2\mu e_{n}H_{0})}$

*. The expressions of the manufacturer’s optimal carbon abatement investment level, optimal production quantity of low-carbon products, and the corresponding excessive abatement level are*

$\tau^{**}=\frac{(1-\mu e_{n})H_{0}}{k+H_{0}-2\lambda H_{0}-2\mu e_{n}H_{0}}$

*,*

$q_{n}^{**}=\frac{k(1-\mu e_{n})}{k+H_{0}-2\lambda H_{0}-2\mu e_{n}H_{0}}$

*, and*

$\tau^{**}-\phi^{**}=\frac{1-\mu e_{n}}{\eta e_{n}}\cdot \frac{2k-H_{0}^{2}+\eta e_{n}H_{0}}{k+H_{0}-2\lambda H_{0}-2\mu e_{n}H_{0}}-\frac{1-t_{0}e_{n}}{\eta e_{n}}$

*, respectively, where*

$1-\mu e_{n}>0$

*and*

$k>(2\lambda +\mu e_{n})H_{0}$

*.*


According to Lemma 2, the effects of the environmental damage coefficient μ, carbon abatement cost coefficient k, consumer low-carbon preference coefficient λ, carbon tax rate *t*_0_, and reward-punishment coefficient *η*on the government’s optimal pre-determined abatement target, the manufacturer’s optimal carbon abatement investment level, and the production quantity of low-carbon products are demonstrated in Propositions 8–11, respectively.

**Proposition** **8.**
*If*

$(2\lambda +\mu e_{n})H_{0}<k<(2\lambda +1)H_{0}$

*, then*

$\frac{\partial \phi^{**}}{\partial \mu }<0$

*,*

$\frac{\partial \tau^{**}}{\partial \mu }>0$

*and*

$\frac{\partial q_{n}^{**}}{\partial \mu }>0$

*; if*

$k>(2\lambda +1)H_{0}$

*, then*

$\frac{\partial \phi^{**}}{\partial \mu }>0$

*,*

$\frac{\partial \tau^{**}}{\partial \mu }<0$

*and*

$\frac{\partial q_{n}^{**}}{\partial \mu }<0$

*.*


Proposition 8 shows that the government imposes a looser emission control policy on the manufacturer with more heavy environmental damage to further promote the carbon abatement investment level and production quantity of low-carbon products when the abatement cost coefficient is relatively low ((2*λ* + *μe_n_)H*_0_ < *k* < (2*λ* + 1)*H*_0_). It is expected to better ensure the maximization of social welfare by increasing the manufacturer’s sales revenue and consumer surplus. Conversely, when the abatement cost coefficient is relatively high (*k* > (2*λ* + 1)*H*_0_), the government imposes a stricter emission control policy on the manufacturer with more severe environmental damage to further reduce the manufacturer’s carbon abatement investment level and production quantity of low-carbon products. At this time, the maximization of social welfare is better ensured by reducing the manufacturer’s carbon abatement cost and environmental damage. To sum up, it can be found that, for manufacturers with higher environmental damage, abatement cost control is more critical. When the abatement cost can be effectively compressed and reduced, the government’s relatively loose emission control policy can maximize social welfare while ensuring the growth of low-carbon product demand and higher incentive income brought by abatement investment activities for the manufacturer. Consequently, a win-win target of economic and environmental benefits is achieved.

**Proposition** **9.**
*If*

$\lambda <\frac{\left(t_{0}-2\mu )e_{n}+2(1-\mu e_{n}\right)}{3}$

*, then*

∂ϕ**∂k<0

*,*

∂τ**∂k<0

*, and*

$\frac{\partial (\tau^{**}-\phi^{**})}{\partial k}>0$

*; if*

$\frac{\left(t_{0}-2\mu )e_{n}+2(1-\mu e_{n}\right)}{3}<\lambda <\frac{\left(t_{0}+\eta -2\mu )e_{n}+2(1-\mu e_{n}\right)}{3}$

*, then*

∂ϕ**∂k<0

*,*

∂τ**∂k<0

*, and*

$\frac{\partial (\tau^{**}-\phi^{**})}{\partial k}<0$

*; if*

$\lambda >\frac{\left(t_{0}+\eta -2\mu )e_{n}+2(1-\mu e_{n}\right)}{3}$

*, then*

$\frac{\partial \phi^{**}}{\partial k}>0$

*,*

∂τ**∂k<0

*, and*

$\frac{\partial (\tau^{**}-\phi^{**})}{\partial k}<0$

*.*


Proposition 9 illustrates that, under the emission control policy of maximizing social welfare, the manufacturer’s carbon abatement investment level is still negatively correlative with the abatement cost coefficient. However, the effect of the abatement cost coefficient on the strictness of the emission control policy and the manufacturer’s excessive abatement level also depends on the consumers’ environmental awareness. When the consumer low-carbon preference is relatively weak ($\lambda <\frac{\left(t_{0}+\eta -2\mu )e_{n}+2(1-\mu e_{n}\right)}{3}$), the government implements a stricter emission control policy as the carbon abatement cost coefficient decreases. Otherwise, a looser emission control policy is issued when the manufacturer’s carbon abatement cost is declining. This also indicates that, under the situation with higher consumer environmental awareness, the government relaxes the carbon abatement requirement for the manufacturer whose abatement cost declines continuously. Consequently, social welfare maximization can be achieved while ensuring the interests of manufacturers, consumers, and environmental benefits. Additionally, for the manufacturer, only when the consumer low-carbon preference coefficient exceeds a certain threshold ($\lambda >\frac{\left(t_{0}-2\mu )e_{n}+2(1-\mu e_{n}\right)}{3}$), the falling abatement cost coefficient can always cause a higher increment in the manufacturer’s carbon abatement investment level compared to the change in the government pre-determined abatement target. Eventually, it will be more helpful for the manufacturer to meet the government’s carbon abatement requirement and achieve a higher excessive abatement level.

**Proposition** **10.**
*If*

$k<\frac{H_{0}^{2}[1+2e_{n}(t_{0}-\mu )]}{2(H_{0}+2\lambda +2\mu e_{n}-1)+\eta e_{n}}$

*, then*

∂ϕ**∂λ>0

*,*

∂τ**∂λ>0

*, and*

$\frac{\partial (\tau^{**}-\phi^{**})}{\partial \lambda }<0$

*; if*

$\frac{H_{0}^{2}[1+2e_{n}(t_{0}-\mu )]}{2(H_{0}+2\lambda +2\mu e_{n}-1)+\eta e_{n}}<k<\frac{H_{0}^{2}[1+2e_{n}(t_{0}+\eta -\mu )]}{2(H_{0}+2\lambda +2\mu e_{n}-1)}$

*, then*

∂ϕ**∂λ>0

*,*

∂τ**∂λ>0

*, and*

$\frac{\partial (\tau^{**}-\phi^{**})}{\partial \lambda }>0$

*; if*

$k>\frac{H_{0}^{2}[1+2e_{n}(t_{0}+\eta -\mu )]}{2(H_{0}+2\lambda +2\mu e_{n}-1)}$

*, then*

$\frac{\partial \phi^{**}}{\partial \lambda }<0$

*,*

∂τ**∂λ>0

*, and*

$\frac{\partial (\tau^{**}-\phi^{**})}{\partial \lambda }>0$

*.*


Proposition 10 indicates that, under the emission control policy of maximizing social welfare, the manufacturer’s carbon abatement investment level is still positively correlative with the consumer low-carbon preference coefficient. However, the effect of the consumers’ environmental awareness on the strictness of the emission control policy and the manufacturer’s excessive abatement level also depends on the abatement cost coefficient. Specifically, when the carbon abatement cost coefficient is relatively high ($k>\frac{H_{0}^{2}[1+2e_{n}(t_{0}+\eta -\mu )]}{2(H_{0}+2\lambda +2\mu e_{n}-1)}$), a stricter emissions control policy is issued if the consumers’ environmental awareness declines continuously. Otherwise, the government often encourages the manufacturer to carry out carbon abatement activities by imposing a loose emission control policy as the low-carbon preference coefficient decreases. This also implies that, only for the manufacturer who can effectively compress and reduce the abatement cost, the government will implement a looser emission control policy in a situation with the weaker consumers’ environmental awareness. Consequently, social welfare maximization can be achieved while promoting the manufacturer’s carbon abatement activities. Additionally, for the manufacturer, when the carbon abatement cost coefficient exceeds a certain threshold ($k>\frac{H_{0}^{2}[1+2e_{n}(t_{0}-\mu )]}{2(H_{0}+2\lambda +2\mu e_{n}-1)+\eta e_{n}}$), the rising low-carbon preference coefficient always helps the manufacturer better meet the government’s carbon abatement requirement and achieve excess abatements.

**Proposition** **11.**

$\frac{\partial \tau^{**}}{\partial t_{0}}>0$

*and*

$\frac{\partial \tau^{**}}{\partial \eta }>0$

*.*


Proposition 11 implies that, different from the situation where the government pre-determined abatement target is given, the manufacturer’s carbon abatement investment level positively correlates with the carbon tax rate and the reward-punishment coefficient under the emission control policy of maximizing social welfare. That is to say, levying a carbon tax and raising the tax rate or adopting a reward-punishment mechanism and raising the reward-punishment coefficient can better promote the manufacturer’s carbon abatement activities. This is mainly because the government’s optimal pre-determined abatement target is closely related to the carbon tax rate and the reward-punishment coefficient. The government can always adjust the strictness of the emission control policy by changing the carbon tax rate and the reward-punishment coefficient, thereby inducing the manufacturer to actively carry out abatement investment activities. This also illustrates that the government can clarify the effect of the carbon tax rate and the reward-punishment coefficient on the manufacturer’s carbon abatement investment level through deciding the optimal pre-determined abatement target, thereby achieving the target of maximizing social welfare.

Lastly, this section further explores the effect of the carbon tax rate and the reward-punishment coefficient on the government’s optimal pre-determined carbon abatement target, the manufacturer’s abatement investment level, and the excessive abatement level. Referring to [39] and combined with an actual situation in practice, we set the base parameters as follows. For instance, parameter *λ* was equal to 0.2 (*λ* = 0.2), which represents consumers’ preference degree for low-carbon products. The carbon abatement cost coefficient was 4.2 (*k* = 4.2), which mainly determines the manufacturer’s one-time low-carbon investment cost. Furthermore, the reward-punishment coefficient was 1.5 (*η* = 1.5), and the carbon tax rate was 0.5 (*t*_0_ = 0.5), both of which to some extent reflect the intensity of the government emission control. Finally, the environmental damage coefficient was set as 0.2 (*μ* = 0.2), and the numerical analysis and graphical visualization were performed using Matlab R2019a. The numerical results are shown in Figure 2 and Figure 3, respectively.

It can be observed from Figure 2 that the higher the manufacturer’s carbon emissions of unit product, the stricter the government’s emission control policy is, which also leads to a lower excessive abatement level. A possible explanation is that the increment in the carbon abatement investment level is lower than that in the pre-determined abatement target caused by the rising carbon emissions of unit product. Thus, this also makes it more difficult for the manufacturer to meet the governments abatement requirement.

Moreover, under the emission control policy of maximizing social welfare, the effect of the carbon tax rate on the government’s optimal pre-determined abatement target is closely related to the manufacturer’s carbon emissions of the unit product. More specifically, when the carbon emissions of the unit product are relatively low (*e_n_* > 1.59), the government’s optimal pre-determined abatement target initially decreases and then increases with the increase of the carbon tax rate. That is to say, within this range, the high carbon tax rate and low pre-determined abatement target are the optimal strategic combination for the government’s mixed emissions control policy when the carbon tax rate is below certain thresholds. Meanwhile, as the carbon emissions of unit product increase, the threshold value of the carbon tax rate that makes the government’s optimal pre-determined abatement target begin to increase becomes lower. This also implies that, in order to effectively control the production quantity of high-emission products, the government is increasingly inclined to a mixed emission control policy with a high carbon tax rate and a high pre-determined abatement target. Until the carbon emissions of the unit product exceed a certain threshold (*e_n_*> 1.59), the rising carbon tax rate will always prompt the government to implement a stricter emission control policy (i.e., higher pre-determined carbon abatement target). Furthermore, within this range, the higher the carbon emissions of the unit product, the greater the increment in the optimal pre-determined abatement target caused by the rising carbon tax rate.

Last but not least, it can be found in Figure 2c that the rising carbon tax rate always leads to a more excessive abatement level, which indicates that the manufacturer can always obtain more incentive income through actively carrying out carbon abatement activities. The reason is mainly because the rising carbon tax rate promotes the continuous and higher increase in the manufacturer’s carbon abatement investment level. Therefore, under the emission control policy of maximizing social welfare, levying a carbon tax or raising the tax rate always positively affects the manufacturer’s carbon abatement investment level and excessive abatement level.

It can be found from Figure 3 that the effects of the carbon emissions of unit product on the government’s optimal pre-determined abatement target, the manufacturer’s abatement investment level, and the excessive abatement level are consistent with the results in Figure 2. However, as shown in Figure 3a, the rising reward-punishment coefficient always enhances the strictness of the emission control policy. Moreover, the increment in the carbon abatement investment level is lower than that in the pre-determined abatement target caused by the rising carbon emissions of unit product. Consequently, with the increase of the reward-punishment coefficient, the excessive abatement level of the manufacturer decline, as shown in Figure 3c. That is to say, although the rising reward-punishment coefficient can improve the manufacturer’s carbon abatement investment level, the stricter emission control policy makes it more difficult for the manufacturer to meet the government’s carbon abatement requirement. Therefore, under the emission control policy of maximizing social welfare, the adjustment effect of the abatement investment level and the excessive abatement level in the same direction cannot be obtained by changing the reward-punishment coefficient.

## 6. Conclusions

In this work, we mainly studied a monopolistic manufacturer’s production and carbon abatement investment decisions in a single period under a mixed emission control policy with carbon tax and a dynamic reward-punishment mechanism for carbon abatements. Firstly, a profit maximization model was formulated, and the effects of the mixed emissions control policy on the manufacturer’s low-carbon operations and the environment were investigated. Then, we constructed a social welfare maximization model to explore the decision-making process of the optimal emission control policy and the corresponding effect on the manufacturer’s carbon abatement behaviors. Finally, through theoretical analysis and numerical analysis, some conclusions, managerial insights, and policy implications were provided as follows:(1)Under a given mixed emission control policy, the abatement cost coefficient is always negatively related to the manufacturer’s carbon abatement investment level and production of low-carbon products. However, only when the consumer low-carbon preference coefficient exceeds a certain threshold, the rising abatement cost coefficient does not result in a higher sales price of low-carbon products and passes on the increased cost to consumers. Additionally, the enhanced consumers’ environmental awareness always effectively promotes the manufacturer to carry out the abatement investment and low-carbon production activities. Meanwhile, only when the carbon abatement cost is relatively low, the manufacturer further benefits consumers by reducing sales prices with the increase of the low-carbon preference coefficient. Furthermore, from the perspective of the environment, the effects of the abatement cost coefficient and the low-carbon preference coefficient are closely related to the strictness of the mixed emission control policy. This indicates that the manufacturer should choose the effective carbon abatement approaches according to the government pre-determined abatement target to reduce environmental damage, such as cutting the abatement cost or enhancing consumers’ environmental awareness. In particular, only when the government pre-determined abatement target is below a certain threshold (*ϕ* < *ϕ*_5_), both of these approaches are beneficial to the manufacturer’s economic and environmental targets.(2)Under a given mixed emissions control policy, a higher pre-determined abatement target does not effectively incentivize the manufacturer to improve the carbon abatement investment level. Instead, it increases the manufacturer’s cost burden and reduces the market demand for low-carbon products. Interestingly, in the situation with a lower carbon abatement cost or a stronger consumer environment awareness, reducing the sales price of low-carbon products is often more viable for the manufacturer when facing a stricter emissions control policy. This also means that both reducing the carbon abatement cost and increasing consumers’ environmental awareness can effectively avoid the excessive transfer of operational costs by the manufacturer under an increasingly stricter emissions control policy. In addition, only when the pre-determined abatement target is greater than the threshold value (*ϕ* > *ϕ*_2_), a stricter emission control policy is beneficial to the reduction of the total carbon emissions.(3)Under a given mixed emission control policy, the relationship between the carbon tax rate or the reward-punishment coefficient and the manufacturer’s abatement investment level and production quantity of low-carbon products mainly depends on the strictness of the emission control policy. Surprisingly, levying a carbon tax or raising the tax rate can effectively reduce the manufacturer’s total carbon emissions, but only when the government pre-determined abatement target is below a certain threshold (*ϕ* = *ϕ*_2_), and adopting a reward-punishment mechanism or raising the reward-punishment coefficient is beneficial to the environment.(4)The government pre-determined abatement target is jointly related to the manufacturer’s carbon abatement cost and consumers’ environmental awareness. For instance, under a situation with higher consumer environmental awareness, the government should impose a looser emission control policy for the manufacturer whose abatement cost declines continuously. At this time, it is also more helpful for the manufacturer to compress and cut the abatement cost to meet the government’s emission control requirements. In addition, under a situation with a lower carbon abatement cost for the manufacturer, a looser emission control policy should also be implemented as consumers’ environmental awareness enhances. Consequently, it is expected that social welfare maximization can be achieved while promoting the manufacturer’s carbon abatement activities.(5)The government pre-determined abatement target is also related to the carbon tax rate and the reward-punishment coefficient. Therefore, the effect of the carbon tax rate on the government’s optimal pre-determined abatement target has a close correlation with the manufacturer’s carbon emissions of the unit product. As the carbon emissions of the unit product increase, the government is increasingly inclined to a mixed emission control policy with a high carbon tax rate and a high pre-determined abatement target. Unlike the carbon tax, the rising reward-punishment coefficient always enhances the strictness of the emission control policy. More interestingly, under the emission control policy of maximizing social welfare, levying a carbon tax and raising the tax rate or adopting a reward-punishment mechanism and raising the reward-punishment coefficient can better promote the manufacturer’s carbon abatement activities. However, considering that the government’s optimal pre-determined abatement target has been raised to varying degrees, changes in the carbon tax rate and the reward-punishment coefficient have the opposite effect on the manufacturer’s excessive abatement level.

Although this study is well-sustained by the literature and integrates manufacturer’s low-carbon operation decisions and government emission control policies, it can be further expanded in the following two aspects. For instance, more incentive-punishment emission control policies can be introduced to model and analyze, such as a reward-punishment-tiered carbon tax policy and dynamic reward-punishment mechanism on carbon emissions. The second possible direction is to extend the profit maximization model formulated in this paper into a multi-product/multi-period setting and explore the effects of different emission control policies on production and carbon abatement investment decisions.

## Figures and Tables

**Figure 1 ijerph-19-13472-f001:**
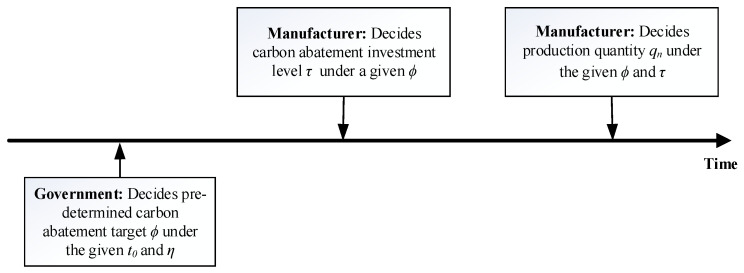
Sequence of participants’ decisions.

**Figure 2 ijerph-19-13472-f002:**
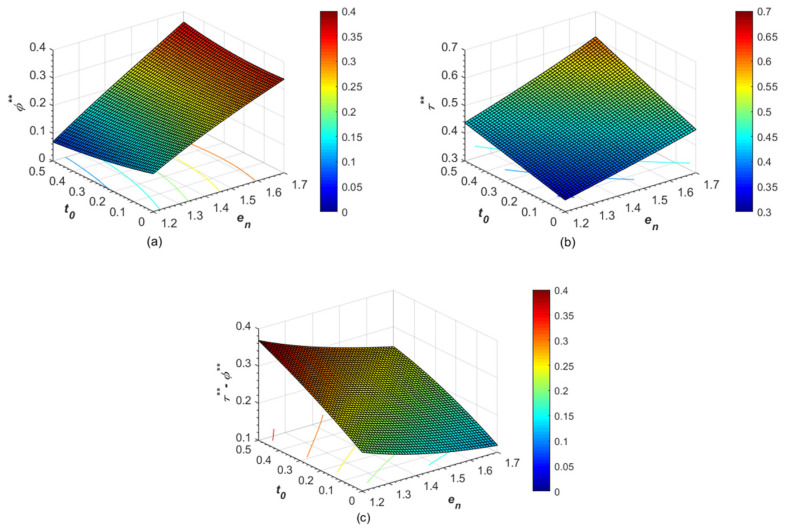
Effects of *e_n_* and *t*_0_ on (**a**) optimal pre-determined abatement target *ϕ***, (**b**) carbon abatement investment level *τ***, and (**c**) excessive abatement level *τ*** − *ϕ***.

**Figure 3 ijerph-19-13472-f003:**
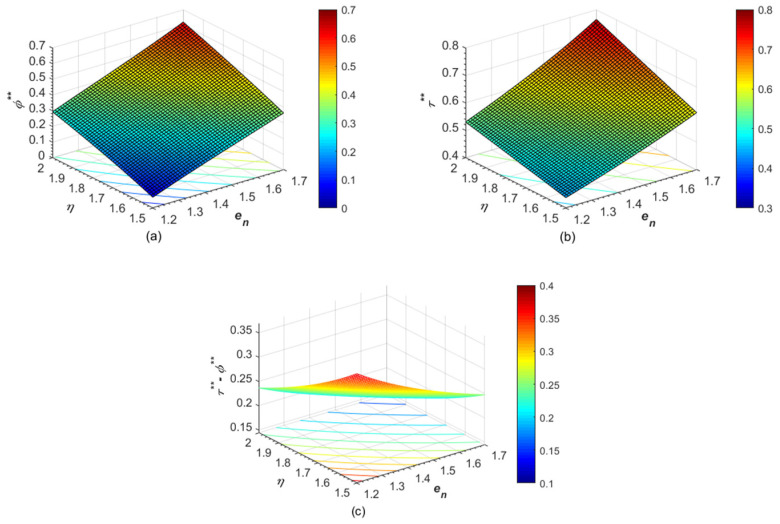
Effects of *e_n_* and *η* on (**a**) optimal pre-determined abatement target *ϕ***, (**b**) carbon abatement investment level *τ***, and (**c**) excessive abatement level *τ*** − *ϕ***.

**Table 1 ijerph-19-13472-t001:** Summary of relevant literature and the position of our paper.

Reference	Carbon Tax	Reward-Punishment Mechanism		Abatement Investment Decision	Consumers’ Environmental Awareness	Decision Makers	
		Based on Carbon Emissions	Based on Carbon Abatements			Manufacturers	Governments
[25,26,27,28,29,30]				✓	✓	✓	
[7,31,33,34]	✓			✓		✓	
[32]	✓			✓		✓	✓
[4,35,36]	✓			✓	✓	✓	
[37]	✓				✓	✓	
[40]		✓		✓		✓	
[41]		✓				✓	
[42,43]		✓				✓	
Our paper	✓		✓	✓	✓	✓	✓ *

* ‘✓’ indicates that a reference includes the corresponding content.

**Table 2 ijerph-19-13472-t002:** Relevant parameters and decision variables.

Decision Variables	Descriptions
*q_n_*	Production quantity of low-carbon products
*τ*	Abatement investment level
**Relevant parameters**	**Descriptions**
*p_n_*	Sales price of unit product
*e_n_*	Carbon emissions of unit product
* *t* _0_ *	Carbon tax rate
*k*	Carbon abatement cost coefficient
*λ*	Consumer low-carbon preference coefficient
*η*	Reward-punishment coefficient
*ϕ*	Government pre-determined emission reduction target, 0 < *ϕ* < 1
*μ*	Environmental damage coefficient
*π_m_*	Manufacturer’s total profit
*E_m_*	Manufacturer’s total carbon emissions
*π_c_*	Consumer surplus
*π_e_*	Environmental damage cost
*f_g_*	Government expenditure
*π_g_*	Social welfare

## Data Availability

The dataset used and/or analyzed in this study is available from the corresponding author on reasonable request.

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
