# Peer review of "Effects of a Mixed Emissions Control Policy on the Manufacturer’s Production and Carbon Abatement Investment Decisions"

_ijerph, 2022, doi:10.3390/ijerph192013472_

Round 1
Reviewer 1 Report
My comments are mentioned in the attached file.

Author Response
Dear editors and reviewers,
Thanks for your letter and reviewer comments concerning our manuscript (ijerph-1940968). Those comments are all valuable and very helpful for revising and improving our manuscript, as well as important guiding significance to our studies. We have studied the comments carefully and have made corrections which we hope you will meet with approval. Revised portions are marked in red in the paper. The main corrections in the paper and the responses to the reviewer’s comments are as follows and also in the attached file:
Reviewer 1#:
- The title is too long, it must be concise and well defined the meaning of research done.
A1: Thanks for your comments. According to your suggestion, we have changed the title of our manuscript to “Effects of a mixed emissions control policy on the manufacturer’s production and carbon abatement investment decisions”. Please see the revised manuscript.
- References are needed for the contents of lines 47-53 on Page 2.
A2: Thanks for your instructive suggestions. According to your comments, we have added the corresponding references for the contents of lines 47-53 on Page 2. Besides, we also added related references for the contents of lines 30-35 on Page 1 and lines 71-73 on Page 2. Please see the revised manuscript.
- Lines 184-186 must be changed. Repetition of words are used here.
A3: Thanks for your careful review of our manuscript. As your comments, the original words have been changed to " First, to the best of our knowledge, this paper was the first study on the reward-punishment mechanism based on the government's pre-determined abatement target and the manufacturer’s actual product low-carbon level." Please see lines 191-193 on Page 4 in the revised manuscript.
- Please define why these parameters are relevant to this study on Page 5?
A4: Thanks for your comments. We are sorry for our inappropriate expression in this sentence. We just want to list all parameters and decision variables involved in our problem models in Table 2. Thus, the original words have been changed to “All parameters and decision variables involved in our models are shown in Table 2.” Please see lines 222-223 on Page 5 in the revised manuscript.
- Authors mentioned previous literature by names of the authors (e.g., Cao et al. etc.). It would be great to alter this repetition.
A5: Thanks for your careful review of our manuscript. According to your suggestion, we have modified similar inappropriate statements in the whole manuscript. Please see the revised manuscript.
- How these figures on Pages 13 and 14 are plotted? methodology and software? It is suggested to use/cite this reference for better understanding about surface plots (https://doi.org/10.1016/j.mineng.2021.107279).
A6: Thanks for your instructive suggestions. We have carefully read the reference you provided and it is very enlightening. For the figures in this manuscript, they are the results of numerical analysis and graphical visualization, and are plotted by using Matlab R2019a, as stated in lines 579-581 on Page 14 of the revised manuscript. In addition, combining another reviewer’s suggestion, we have added color changes according to the intensity levels to make the figures more intuitive. Furthermore, the research methods of our manuscript have also been added in Subsection 3.3 to make this study clear. Please see the revised manuscript.
- The conclusions section is very lengthy-must be precise and well written.
A7: Thanks for your valuable advice. We have carefully reorganized and rewritten the conclusions section based on the refined results and discussions. Please see Section 6 in the revised manuscript.

Reviewer 2 Report
Comment:
In this manuscript, the authors perform a comprehensive study on a mixed emissions control policy linked the manufacturer’s low-carbon production with customer’s carbon tax. The results show that the reward system would be effective with certain conditions on both manufacturer side and customer side while the excessive abatement levels of the manufacturers would be limited. However, there are remaining questions on the sensitivity analysis and figure legends. Thus, I believe a minor revision is needed for the manuscript at this stage.
Specific comments:
1. In Figure 2 and 3, the intensity levels with the color change are missed.
2. The authors provide several sensitivity analyses with different factors. However, the quantitative expressions are not discussed in this manuscript. How the values are chosen in Figure 2 and 3?
3. The authors emphasize that the negative effect of the excessive abatement levels of the manufacturers. Is there a clear assessment of the initial excessive abatement levels of the manufacturers for reference?
Author Response
Dear editors and reviewers,
Thanks for your letter and reviewer comments concerning our manuscript (ijerph-1940968). Those comments are all valuable and very helpful for revising and improving our manuscript, as well as important guiding significance to our studies. We have studied the comments carefully and have made corrections which we hope you will meet with approval. Revised portions are marked in red in the paper. The main corrections in the paper and the responses to the reviewer’s comments are as follows and also in the attached file:
Reviewer 2#: In this manuscript, the authors perform a comprehensive study on a mixed emissions control policy linked the manufacturer’s low-carbon production with customer’s carbon tax. The results show that the reward system would be effective with certain conditions on both manufacturer side and customer side while the excessive abatement levels of the manufacturers would be limited. However, there are remaining questions on the sensitivity analysis and figure legends. Thus, I believe a minor revision is needed for the manuscript at this stage.
Specific comments:
- In Figure 2 and 3, the intensity levels with the color change are missed.
A1: Thanks for your instructive suggestions. According to your suggestion, we have added color changes according to the intensity levels to make the figures more intuitive. Please see the revised manuscript.
- The authors provide several sensitivity analyses with different factors. However, the quantitative expressions are not discussed in this manuscript. How the values are chosen in Figure 2 and 3?
A2: Thanks for valuable advice. It is really difficult to get real values from some enterprises. But we tried to investigate some manufacturers in China and Denmark. The parameters are set based on these enterprises’ actual situation in practice. The purpose of these values is just for investigating the effects of the carbon tax rate and the reward-punishment coefficient on the government’s optimal pre-determined carbon abatement target, the manufacturer’s abatement investment level and excessive abatement level. Based on your constructive suggestions, we have thought carefully and explained how some parameters are set in detail and the corresponding economic meaning, please see lines 569-580 in the revised manuscript.
- The authors emphasize that the negative effect of the excessive abatement levels of the manufacturers. Is there a clear assessment of the initial excessive abatement levels of the manufacturers for reference?
A3: Thanks for your comments. We are sorry for your misunderstanding due to our unclear statements. In our manuscript, the excessive abatement level is equal to the difference of the manufacturer’s optimal carbon abatement investment level and the government’s optimal pre-determined abatement target. This indicator represents the degree to which the manufacturer is meeting the government's carbon abatement requirements. If it is positive, the manufacturer will obtain the incentive benefits for excessive carbon abatements, otherwise, a fine is imposed. Please see lines 463-469 on Page 11 in the revised manuscript. Therefore, a higher excessive carbon abatement level is generally positive to the manufacturer and the government. In this paper, we mainly focus on the effect of a mixed emission control policy on the manufacturer’s excessive abatement level, and how to guide manufacturers to achieve the higher excessive abatement levels. These are mainly discussed in Figures 2 and 3. Please see the revised manuscript.

Reviewer 3 Report
Wang and Zhang study on the reward-punishment mechanism based on the government’s pre-determined abatement target and the manufacturer’s actual carbon abatement. In the study, two models were constructed, profit maximization model and social welfare maximization model. The manuscript is well-organized, and conclusions are well supported. Detailed calculation and definition are offered as well. A few minor questions should be addressed.
1. The authors claim “stronger environment awareness of consumers, a stricter emissions control policy can instead cause a reduction in the sales price of low-carbon products”, and “consumers have higher environmental awareness, the government should impose a looser emissions control policy”. (Page 15, Line 616, 637). This seems contradictory. How dose environmental awareness impact policy? What policy should take under high environmental awareness condition, stricter or looser?
2. Some language needs more polish. e.g., it is better to rewrite the sentence of “It is should be noted ….” In line 211 of Page 5 to “It should be note…” or “It is worth noting…”
Author Response
Dear editors and reviewers,
Thanks for your letter and reviewer comments concerning our manuscript (ijerph-1940968). Those comments are all valuable and very helpful for revising and improving our manuscript, as well as important guiding significance to our studies. We have studied the comments carefully and have made corrections which we hope you will meet with approval. Revised portions are marked in red in the paper. The main corrections in the paper and the responses to the reviewer’s comments are as follows and also in the attached file:
Reviewer 3#: Wang and Zhang study on the reward-punishment mechanism based on the government’s pre-determined abatement target and the manufacturer’s actual carbon abatement. In the study, two models were constructed, profit maximization model and social welfare maximization model. The manuscript is well-organized, and conclusions are well supported. Detailed calculation and definition are offered as well. A few minor questions should be addressed.
- The authors claim “stronger environment awareness of consumers, a stricter emissions control policy can instead cause a reduction in the sales price of low-carbon products”, and “consumers have higher environmental awareness, the government should impose a looser emissions control policy”. (Page 15, Line 616, 637). This seems contradictory. How dose environmental awareness impact policy? What policy should take under high environmental awareness condition, stricter or looser?
A1: Thanks for your careful review of our manuscript. We are sorry for your misunderstanding due to our unclear statements. First, the former conclusion (Page 17, Line 666 in the revised manuscript) you mentioned is obtained from Proposition 2. The purpose of this conclusion is to illustrate the reaction of the manufacturer when facing a change in a given emissions control policy. At this time, the objective function is profit maximization of the manufacturer. However, the latter conclusion (Page 17, Line 685 in the revised manuscript) you mentioned is obtained from Proposition 9 and Proposition 10. The purpose of this conclusion is to illustrate the reaction of the government according to the manufacturer’s decision feedback. At this time, the objective function is the social welfare maximization of the government. Thus, these two conclusions are drawn from different decision-makers’ perspectives. To make it easier for the reader to understand, we have carefully reorganized and rewritten the conclusions section based on the refined results and discussions. Please see the revised manuscript.
Moreover, as stated in line 456 on page 11 of the revised manuscript, the government designs and adjusts the emissions control policy according to the manufacturer’s decision feedback. Meanwhile, the manufacturer’s operational decisions are closely related to consumers’ environmental awareness. Thus, consumers' environmental awareness will subsequently affect the design and optimization of the emissions control policy.
Finally, as stated in line 685 on page 17 of the revised manuscript, under a situation with higher consumers’ environmental awareness, the government should impose a looser emissions control policy for the manufacturer whose abatement cost declines continuously. That is to say, the government's pre-determined abatement target is jointly related to the manufacturer’s carbon abatement cost and consumers’ environmental awareness. Please see the revised manuscript.
- Some language needs more polish. e.g., it is better to rewrite the sentence of “It is should be noted ….” In line 211 of Page 5 to “It should be note…” or “It is worth noting…”
A2: Thank you very much to point out the grammatical issues in our manuscript. According to the comments from you and the editors, we have double checked the language by ourselves and polished the manuscript with the professional assistance in writing, conscientiously. Please see the revised manuscript.
